# Be Cool: A Holistic and Innovative Approach to Rehabilitation in Multiple Sclerosis: Study Protocol for a Randomized Controlled Trial

**DOI:** 10.3390/healthcare12090870

**Published:** 2024-04-23

**Authors:** Antonia Kaltsatou, Sofia Theodorou, Anastasios Orologas

**Affiliations:** 1FAME Laboratory, School of Physical Education and Sport Science, University of Thessaly, 42130 Trikala, Greece; 2Greek Multiple Sclerosis Society, 42100 Thessaloniki, Greece; sofiatheodorou@gmss.gr (S.T.); orologas@auth.gr (A.O.)

**Keywords:** exercise training, cooling therapy, psychological support, quality of life, symptom management

## Abstract

(1) Background: Individuals with multiple sclerosis (MS) have to deal with numerous symptoms that adversely impact their quality of life. While pharmaceutical treatments offer some relief, they often fall short of addressing the full spectrum of MS symptoms. To bridge this gap, we introduce the Be Cool rehabilitation program, a comprehensive protocol designed to enhance the well-being and life quality of MS individuals. (2) Methods: The Be Cool program is a multifaceted approach that combines exercise training, nutritional guidance, psychological support, and cooling strategies. Adapted to meet the unique needs of MS individuals, this program aims to mitigate symptoms, promote physical and mental health, and improve overall quality of life. The integration of these strategies addresses the complex challenges faced by MS individuals, offering a holistic solution beyond conventional medication. (3) Conclusions: The Be Cool rehabilitation protocol is designed to offer individuals with MS a comprehensive approach to symptom management, fostering improvements in their quality of life. By addressing the multifaceted nature of MS through an integrated strategy, the program holds promise for more effective management of the condition.

## 1. Introduction


*“Our body has the ability to heal itself. Diet, exercise, environment, lifestyle and way of thinking are of utmost importance.”*
Hippocrates 460–377 BC

Multiple sclerosis (MS) is a chronic, debilitating disease of the central nervous system, predominantly affecting young adults [1]. This disease progressively impairs mobility and functionality, making everyday activities, community engagement, and personal care difficult for those diagnosed with MS. Although pharmaceutical treatments are essential for managing the condition, they may not address all symptoms, potentially affecting the individual’s quality of life [2]. People living with MS often deal with a spectrum of motor and psychological symptoms, which presents a complex challenge in terms of management [3]. While medications can mitigate some symptoms, they may worsen others, underscoring the necessity for a comprehensive and holistic treatment approach. To address this complexity, a comprehensive and multidisciplinary rehabilitation strategy is essential for enhancing the quality of life of individuals with MS.

Previous evidence has proved that regular physical activity is essential in managing MS [4,5]. Exercise not only helps maintain physical strength and endurance but also contributes to better balance and coordination, potentially reducing the risk of falls [6]. Moreover, it can have a positive impact on fatigue and fatigability, which are the most common and debilitating symptoms of MS [7]. Adapted exercise programs, considering individual capabilities and limitations, can significantly enhance the quality of life for MS individuals [8,9]. Even Uthoff’s syndrome, characterized by a temporary worsening of MS-related neurological symptoms due to increased core body temperature, may be managed effectively through exercise combined with cooling strategies [10,11]. These strategies, when applied before or during exercise, can prevent a rise in both core and skin temperatures, thereby enhancing the safety and effectiveness of exercise programs for individuals with MS. Research has demonstrated that such cooling strategies can significantly mitigate the risk of symptom exacerbation, enabling a more comfortable and beneficial physical activity experience for those affected by MS [12]. This form of therapy can be an essential component of symptom management, especially for those who experience a significant worsening of symptoms with heat.

Another critical component for overall health and well-being is balanced nutrition, particularly for individuals with MS. A strong nutritional foundation is essential in managing symptoms, mitigating inflammation, and supporting a robust immune system for those affected by MS. Individualized nutritional strategies that emphasize whole foods, anti-inflammatory components, and sufficient hydration are essential [13]. Malnutrition, often a concern among MS individuals, is closely associated with increased fatigue. Incorporating a diet rich in essential nutrients, such as vitamin D, biotin, and other vital vitamins and minerals, can significantly alleviate fatigue and aid in managing the challenging symptoms of MS [14]. However, there is a pressing need for educational initiatives to guide individuals with MS on the specific dietary choices that best support their unique health requirements, providing insight into beneficial ingredients and optimal nutritional practices.

Psychological support is essential for the mental health and well-being of individuals with MS, in addition to exercise, cooling strategy, and nutrition. Group counsellinginterventions provide a unique opportunity for people living with MS to receive emotional support and share everyday experiences and coping strategies [15,16]. This kind of psychological support can have profound effects on well-being, as it addresses both the psychological and emotional challenges that come with MS [17]. Psychological interventions, such as mindfulness, cognitive, behavioural therapy, and self-help groups, promote a range of positive psycho-social outcomes, such as reducing feelings of isolation, providing valuable information, and offering encouragement from those who truly understand the daily challenges faced by those living with MS [18,19]. A well-rounded rehabilitation protocol that integrates exercise, nutritional guidance, group psychological support, and a cooling strategy can offer a more effective way to manage the myriad symptoms associated with MS. This holistic approach acknowledges the diverse needs of individuals living with MS, aiming to provide relief not just from physical symptoms but also to support mental and emotional well-being. Such a protocol not only helps in symptom management but also contributes to overall life satisfaction, wellness, resilience, and quality of life for those battling this challenging condition.

In this light, we integrate these four components into a holistic combined rehabilitation protocol, which offers a comprehensive approach to managing MS symptoms. This holistic strategy addresses not only the physical aspects of the disease but also the emotional, psychological, and nutritional needs, thereby aiming to improve the overall quality of life for individuals with MS. The collaboration of healthcare professionals from various fields, including neurology, clinical exercise physiology, nutrition, and psychology, is vital in designing and implementing an effective rehabilitation plan. We hypothesize that compared to the control arm (usual care), Be Cool participants will show more significant improvements in the primary outcome of health-related quality of life (QoL) and in the secondary outcomes of functional ability, thermoregulation, nutritional status, mood, well-being, and psychological resilience.

## 2. Materials and Methods

The protocol received ethical approval from the ethics committee of the Greek Multiple Sclerosis Society (9 May 2023, internal ref: 396/23) and is registered at ClinicalTrials.gov: NCT06248281. The participants are being recruited from the Greek Multiple Sclerosis Society (https://gmss.gr/, accessed on 22 January 2024) (GMSS) via community outreach.

### 2.1. Eligibility Criteria

Prospective participants are eligible for the study if they meet the following criteria: they have a confirmed diagnosis of MS according to the McDonald criteria [20], exhibit the relapsing-remitting MS (RRMS) phenotype characterized by clearly defined attacks of new or increasing neurological symptoms, have experienced no relapses in the past six months, are between the ages of 20 and 60 years, have a confirmed diagnosis of MS for at least two years, and exhibit mild or moderate neurological disability, as defined by scoring between 0 to 5.5 on the Expanded Disability Status Scale (EDSS), indicating their ability to walk independently for at least 100 m without a cane [21].

The study will not include participants who have undergone psychotherapy in the past six months. Additionally, those with severe suicidality, including ideation, plan, and intent, or who have experienced one or more relapses in the previous month, have undergone corticosteroid treatment in the last month, have other severe medical conditions in addition to MS, or are currently pregnant, will not be able to participate in the study. After baseline testing, the principal investigator will perform randomization using sealed envelopes to allocate participants in a 1:1 ratio to either the BeCool or Control group.

### 2.2. Recruitment and Trial Procedures

A promotional flyer, providing an overview of the program and contact information, was disseminated through social media channels such as Facebook, Instagram, and Twitter, as well as via email and to members of the Greek MS Society. Interested individuals were then contacted by either the Principal Investigator or representatives from the Greek MS Society’s center. Those meeting the study’s inclusion criteria were invited for a consultation, during which the study’s objectives and methodologies were explained, and a questionnaire was administered to confirm eligibility. Participants who consented to join the study attended a preliminary session for a comprehensive evaluation by a neurologist and to familiarize themselves with the experimental procedures. Subsequently, each participant was scheduled for an experimental session, which lasted for 120 min. To ensure a comprehensive evaluation of the intervention’s impact, a team of professionals from different disciplines, including clinical exercise physiology, nutrition, psychology, and environmental physiology, conduct the assessments. Enrolled participants are encouraged to continue with their usual care, which may vary based on factors such as disability status and disease severity. The usual care often includes services such as disease-modifying treatments, various aids, and physical or occupational therapy, all of which are essential for managing the condition and ensuring optimal support. However, to prevent potential confounding effects on the study outcomes, participants are advised to refrain from participating in structured exercise programs or psychological therapy during the intervention period. This ensures that any changes observed in the study are attributed specifically to the intervention being tested rather than to additional interventions that participants may undergo concurrently. Before, after three months, and after the six months of the Be Cool program, all the participants will attend an experimental session where they will undergo the following assessments.

### 2.3. Physiological Measurements

Core temperature (index by visceral temperature) will be recorded continuously using a non-harmful telemetric capsule (e-Celsius Performance, BodyCap, Caen, France). Skin temperatures will be continuously recorded at four sites using iButton temperature sensors (DS1921 H, Maxim/Dallas Semiconductor Corp., Dallas, TX, USA). Specifically, skin temperature will be measured at the upper arm, chest, thigh, and calf, which will be used to estimate mean skin temperature (upper arm: 30%, chest: 30%, thigh: 20%, and calf: 20%). Finally, heart rate will continuously be recorded via a Polar S810 (Kempele, Finland).

### 2.4. Functional Ability Tests

A battery of five functional ability tests will be performed, these are as follows:(1)Two-minute walk test. This test requires that all patients walk as far as possible for over 2 min. It is used to assess an individual’s ability to perform activities of daily living independently [22].(2)Timed 25-Foot Walk (T25-FW) test. The test assesses walking speed achieved over a 25-foot distance. It is a validated test that reflects the patient’s mobility and leg function performance [23].(3)Sit-to-stand test (STS). Participants are encouraged to complete five successive full stands from a seated position. The test is used to indicate lower limb strength, balance, and mobility [18,19]. It has been reported that the STS times are associated with standing and leaning balance and mobility in older people. Slow STS times have also been found to predict subsequent disability, falls, and hip fractures [24].(4)Berg Balance Scale (BBS). The BBS evaluates the performance in 14 specific activities that require a balance function (e.g., picking up an object or standing on one leg) necessary for daily living activities. Participants are scored on a 5-point (0–4) ordinal scale depending on their ability to complete the requested actions. A score of 0 is assigned when the task cannot be achieved, and a score of 4 indicates independence. A score of 45 or less indicates a high risk for falls [25].(5)The Mini-BESTest (Balance Evaluation Systems Test) is a streamlined clinical assessment tool designed to evaluate balance impairments in individuals with various conditions, including neurological disorders. It is a condensed version of the original BESTest, aiming to provide a more practical and time-efficient method for clinicians to assess patients’ balance function [26].

### 2.5. Nutrition Assessment

The effectiveness of nutritional education sessions will be conducted by implementing dietary assessment tools and questionnaires. This provides valuable insights into participants’ dietary patterns, nutrient intake, and lifestyle habits. The questionnaires that will be used are provided below:The 7-day food diary is a detailed record where participants log everything they eat and drink over a week. This diary approach allows for a comprehensive analysis of an individual’s food and nutrient intake, capturing day-to-day variations in eating habits.The self-administered, semi-quantitative food frequency questionnaire (FFQ) of 36 items (medium-FFQ); the medium-FFQ is a self-administered questionnaire designed to estimate an individual’s habitual food and nutrient intake over the previous 2–3 months.The 12-item questionnaire on dietary and lifestyle habits. After each biweekly nutrition education session, participants are asked to complete a 12-item questionnaire focusing on their dietary and lifestyle habits during the last week.

### 2.6. Psychological Intervention Evaluation

To evaluate the effectiveness of the psychoeducational program, the following established questionnaires will be used:The Hospital Anxiety and Depression Scale—HADS: The HADS is a widely used instrument designed to detect states of depression and anxiety. It consists of 14 items, divided equally into two subscales: one for anxiety and the other for depression.Perceived Stress Scale—PSS: The PSS measures the degree to which situations in one’s life are appraised as stressful. It is a self-reported questionnaire that assesses the perception of stress over the past month. Items are designed to understand how unpredictable, uncontrollable, and overloaded respondents find their lives.Cognitive Fusion Questionnaire—CFQ-7: The CFQ-7 is a short form of the original Cognitive Fusion Questionnaire. It assesses the extent to which an individual experiences fusion with their thoughts or the degree to which they identify with their thoughts so closely that they are unable to separate themselves from them.Valued Living Questionnaire—VLQ: The VLQ measures the extent to which individuals live according to their values. It consists of two parts: the first part assesses the importance of different value domains (e.g., family, health, education), and the second part evaluates how consistently individuals act in accordance with these values.MPFI Psychological Flexibility subscale: The MPFI assesses an individual’s ability to be in the present moment with full awareness and openness to experiences and to take action guided by their values. This construct is central to acceptance and commitment therapy and is associated with better mental health outcomes.Connor–Davidson Resilience Scale—CDRISC-25: The CD-RISC-25 is a 25-item scale measuring the strength of an individual’s resilience, or their ability to cope with and bounce back from adversity. The scale covers various aspects of resilience, including personal qualities, the positive acceptance of change, and secure relationships.Multiple Sclerosis Quality of Life-54 (MSQOL-54): The MSQOL-54 is an MS-specific quality-of-life instrument that includes generic and MS-specific items. It encompasses 54 items covering a broad range of physical, mental, and social health domains, providing a comprehensive assessment of the quality of life in MS individuals.The patient-determined disability scale (PDDS): The PDDS serves as a valuable self-reported outcome measure for assessing disability. This scale allows patients to provide their perspectives on the impact of MS on their daily functioning, reflecting their unique experiences and challenges [27].

### 2.7. Intervention

This program is structured to provide a holistic, supportive, and adaptable approach to managing MS, focusing on physical health, nutritional balance, psychological support, and practical strategies for daily living (Figure 1). The program includes:

#### 2.7.1. Exercise Intervention and Education

The exercise program included in this intervention has been specifically designed to encourage participants to increase physical activity while reducing sedentary behaviour, thereby promoting behavioural change. Participants will engage in structured exercise sessions that are tailored to promote activity behaviour change, which may include a variety of exercises such as aerobic activities (e.g., walking, cycling), strength training, flexibility exercises, and balance training, depending on the participant’s goals and capabilities. An exercise physiologist will provide regular support and guidance to ensure that participants are on track. The exercise program will include:

Education Sessions: Participants will meet virtually with an exercise physiologist every two weeks via Microsoft Teams 24074.2321.2810.3500. During these sessions, participants will receive personalized education about their specific needs, including exercise techniques, goal setting, overcoming barriers to physical activity, and strategies for increasing daily movement and reducing sedentary time. Furthermore, the exercise physiologist will discuss the benefits of regular exercise. These benefits may include improved cardiovascular health, enhanced muscular strength and endurance, better flexibility and balance, increased energy levels, stress reduction, mood enhancement, weight management, and overall improvement in quality of life. Understanding these benefits will help participants appreciate the importance of incorporating exercise into their daily routine and motivate them to adhere to the program.

Exercise Videos: Weekly, customized exercise videos will be uploaded to the Teams platform for participants to access. These videos will provide guidance on performing exercises safely and effectively, allowing participants to engage in physical activity at their convenience while receiving ongoing support and instruction.

Mobile App Integration: A mobile app will be developed to track and monitor participants’ daily exercise levels. The app will be integrated with Google Health Fit and iOS Health Fit, allowing participants to monitor their progress, set goals, receive reminders easily, and access resources to support their physical activity progress.

The exercise program will be tailored to each participant’s needs, abilities, and preferences. Modifications may be made based on factors such as fitness level, health status, mobility limitations, and personal goals. Throughout the intervention period, participants’ progress will be closely monitored through functional ability tests, including the 2-min walk test, the five-times sit-to-stand test, etc. These tests will be conducted every three weeks to assess participants’ functional abilities and overall improvement. Based on the results of these functional ability tests and ongoing monitoring of participants’ progress, the intensity and content of the exercise program will be adjusted accordingly. If participants demonstrate improvements in their functional ability and overall fitness levels, the exercise physiologists will modify the program to gradually increase the intensity or introduce new exercises to continue challenging them. Conversely, if participants experience difficulties or limitations during the program, modifications will be made to ensure their safety and comfort while promoting progress toward their goals. These modifications may include reducing the intensity of certain exercises, focusing on specific areas of weakness or discomfort, or providing alternative exercises that better suit the individual’s needs.

#### 2.7.2. Nutritional Evaluation and Education

Nutrition education within the intervention is based on evidence-based principles of maintaining a balanced diet to support the specific needs of individuals with the condition. The guidance provided by the nutritionist is tailored to each participant’s dietary requirements, taking into account factors such as their medical history, nutritional needs, and personal preferences. Changes in participants’ dietary habits and adherence to nutritional recommendations are measured through various methods. These may include dietary assessments such as food diaries or 24-h recall interviews to track participants’ food intake and identify any changes over time. Additionally, participants’ weight, body composition, and other relevant health markers may be monitored to assess the impact of dietary changes on their overall health and well-being. A nutritious recipe is uploaded to the platform weekly to further support participants in adopting healthy eating habits. These recipes not only provide practical guidance on preparing nutritious meals but also serve as a source of inspiration and motivation for participants to incorporate healthier food choices into their daily lives.

Overall, the nutrition education component of the intervention aims to empower participants with the knowledge and skills they need to make informed dietary choices and optimize their nutritional intake to support their condition and overall health. Regular meetings with the nutritionist and access to practical resources such as nutritious recipes enhance participants’ ability to adhere to dietary recommendations and achieve their nutritional goals.

#### 2.7.3. Psychological Intervention

In the psychological component of the intervention, group organization is carefully tailored to ensure participants benefit optimally from shared experiences and support. Led by a psychologist, weekly group meetings conducted via Microsoft Teams focus on addressing participants’ psychological well-being. These sessions utilize a combination of acceptance and commitment therapy (ACT) techniques rooted in the tradition of behavioural therapy. Group organization may be based on factors such as participant preference or similarity in MS symptoms and challenges. ACT emphasizes psychological flexibility as a critical process for improving mental health, with core components including cognitive defusion, acceptance, self as context, values clarification, and committed action. Additionally, participants receive supplementary psychoeducational reading material aimed at enhancing coping strategies for managing MS challenges. This material provides valuable insights and practical tools to help participants manage MS more effectively. By integrating group therapy sessions with ACT techniques and psychoeducational resources, participants are equipped with a comprehensive toolkit to manage MS symptoms, negotiate challenges, and improve their overall quality of life [28].

#### 2.7.4. Head and Neck Cooling Strategy

The cooling strategy is designed to address temperature regulation in MS and provides symptomatic relief for participants experiencing thermo-sensitivity. Participants will be advised to apply a cooling cap (Headcool Power, Inuteq) to the head and neck area for a minimum of two hours daily. This cooling cap has been chosen for its effectiveness in providing targeted cooling and alleviating MS-related symptoms associated with heat sensitivity. While the study does not specify a maximum duration for the cooling strategy, the minimum duration of two hours daily is recommended to achieve optimal benefits. Regarding temperature, the cooling cap is designed to maintain a consistent cooling temperature within a comfortable range suitable for individuals with MS. The specific time of day when the cooling will be applied is not critical. However, participants will be advised to use the cooling method during their exercise sessions, when engaged in outdoor activities, and any time they do physical work that may increase their body temperature.

### 2.8. Comprehensive Assessment

To monitor progress and adjust the program as needed, participants’ functional ability, nutritional status, psychological condition, and overall quality of life will be evaluated at three key points: baseline (before starting the program), after three months, and upon completion of the program at six months.

### 2.9. Control Group—Usual Care

Participants in the control group will follow the usual care. Usual care refers to the standard care participants would receive if they were not involved in the study. This includes routine treatments and services offered in the participant’s usual healthcare setting. Participants will continue to receive their current treatment, if necessary, and may also receive additional treatments if recommended by their general practitioner or specialist due to any comorbidities. Throughout the program, participants will be encouraged to elevate their habitual levels of exercise. This encouragement will be facilitated by the utilization of a newly developed application designed to estimate participants’ exercise levels through activity tracking.

### 2.10. Blinding

Randomization will be blinded to outcome assessors and the statistician. A blinded interpretation of the study results will also be conducted [29].

### 2.11. Satisfaction with the Intervention

A satisfaction questionnaire consisting of 10 questions will be sent every week to evaluate participant satisfaction with the Be Cool interventions.

### 2.12. Adherence to the Interventions

Each week, the implementers will gather attendance information from participants and ask them to rate their commitment to homework activities.

## 3. Data Analysis

### 3.1. Sample Size Calculation

Our study evaluates the effects of a holistic rehabilitation program on individuals with MS, utilizing key metrics to determine improvements in functional ability, nutritional status, and psychological well-being. Determining the minimum required sample size for our protocol was guided by preliminary data on quality of life [30]. Sample size calculations were performed using G*Power 3.0 [31], employing the “ANOVA: Repeated measures, within-between interaction” approach. This method facilitated estimating the study’s power to detect between-group differences and within-subject changes over time, considering three measurement points (baseline, mid-intervention, and post-intervention) for our MS participants. We set the statistical power and α error probability at 0.90 and 0.05, respectively, aligning with rigorous research standards to ensure sufficient sensitivity in detecting meaningful effects. Our calculations indicate that each group requires 15 individuals to power the study sufficiently. Analyses will adhere to the intention-to-treat principle, with multiple imputations used to handle missing data. A per-protocol and sensitivity analysis will also be conducted, excluding participants with significant non-compliance.

Detailed documentation and analysis of baseline demographics, including age, gender, disease duration, and relevant medical history, will provide valuable insights into the characteristics of our study population and allow us to control for potential confounding variables in our analyses. Additionally, thorough documentation of participants’ current disease-modifying therapy (DMT) regimens, including the type of medication, dosage, and duration of treatment, will enable us to assess the impact of DMT use on intervention outcomes and explore potential interactions with our intervention. We aim to conduct subgroup analyses to investigate differences in treatment effects based on baseline demographics and DMT use, with the goal of informing personalized approaches to MS management. By implementing these strategies, we aim to enhance the validity and generalizability of our findings and ultimately improve the care and outcomes of individuals living with MS.

### 3.2. Dissemination and Protocol Amendments

The results of our study, irrespective of their alignment with the initial hypothesis, will be shared transparently across various platforms, including scientific journals, conferences, news outlets, social media, and both digital and print educational resources. We aim to tailor our communication to suit each audience, from in-depth scientific discussions to accessible summaries for the general public. Authorship will be determined based on the ICMJE guidelines, ensuring that all contributors meeting these criteria have access to the data and are involved in its interpretation. We will actively engage with our audience through interactive sessions and feedback mechanisms, ensuring our findings are accessible, inclusive, and ethically communicated. Additionally, we commit to archiving our data and conclusions for long-term access and continuous evaluation of our work’s impact and relevance. This approach to dissemination aims to maximize the reach and utility of our findings, fostering a culture of openness and dialogue in the scientific community and beyond.

## 4. Discussion

The proposed holistic rehabilitation program for individuals with MS is designed with the overarching goal of enhancing health-related quality of life. This multifaceted approach aims to address the complex and varied needs of MS individuals by focusing on several key areas: improving functional ability, optimizing nutritional status, facilitating participation in physical activities through effective thermoregulation strategies, and bolstering resilience and psychological well-being. Each component plays a critical role in the program’s overall success, contributing to a comprehensive plan that acknowledges the interplay between physical health and mental resilience in managing MS.

The holistic rehabilitation program represents a comprehensive approach to managing MS, acknowledging that improvements in physical health are intrinsically linked to mental and emotional well-being. By addressing the multifaceted needs of individuals with MS, the program aims to improve specific symptoms and enhance overall quality of life. The success of such a program lies in its ability to be tailored to the individual needs of participants, reflecting the variability of MS symptoms and their impact on daily life.

Future research should focus on quantifying the outcomes of such holistic interventions and exploring the synergistic effects of combining physical, nutritional, and psychological strategies. Long-term follow-up will be crucial in understanding the sustainability of benefits and the potential for such programs to alter the trajectory of MS progression.

## 5. Conclusions

In conclusion, the proposed holistic rehabilitation program offers a promising pathway to improving the health-related quality of life for individuals with MS. By fostering improvements in functional ability, nutritional status, physical activity participation, and psychological resilience, the program embodies a comprehensive approach to managing the complex and varied challenges of MS. This approach represents a valuable strategy that all healthcare systems ought to offer to individuals diagnosed with chronic conditions.

## Figures and Tables

**Figure 1 healthcare-12-00870-f001:**
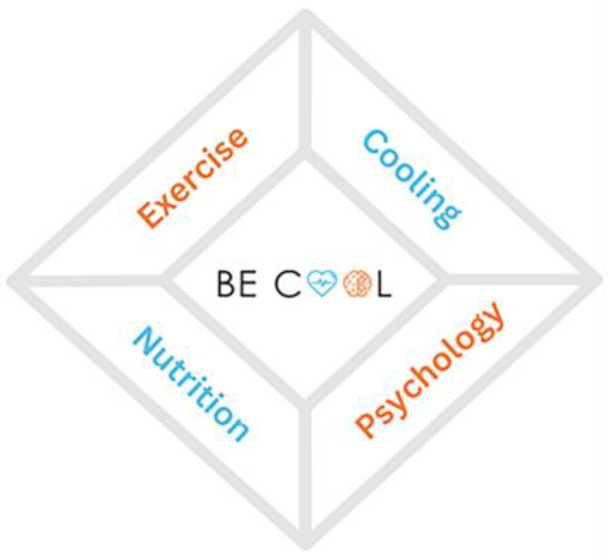
The structure of Be Cool’s rehabilitation program.

## Data Availability

The data that support the findings of this protocol are available from the corresponding author upon reasonable request.

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
