# Peer review of "Be Cool: A Holistic and Innovative Approach to Rehabilitation in Multiple Sclerosis: Study Protocol for a Randomized Controlled Trial"

_healthcare, 2024, doi:10.3390/healthcare12090870_

Round 1

Reviewer 1 Report

Comments and Suggestions for Authors

Thank you very much for giving me the opportunity to review this work. Below I present some questions with the aim of helping to improve the presentation of the work carried out.

CONSIDERATIONS FOR AUTHORS

Line 281-294. The sample size to be included in the study is not specified.

Which professional/s carries out the assessment, and how many assessment participate?

Line 264. Intervention Control Group. The control group continues with the usual treatments, does the intervention group also or are they suspended?

What do exercise programs include and what recommendations are given to participants? It would be convenient to know the program that the participants must follow and if modifications are made, based on what?

Nutrition education, what is it based on? How are changes measured?

In Psychotherapy, if the approaches are done in groups, how are the groups organized?

The cooling strategy, how is it designed? How long maximum and how much minimum? What temperature? Does it matter if it is applied in the morning or afternoon?

Who makes, and how, the distribution of patients to the control group or intervention group?

Thanks you,

Author Response

Point-to-Point response to the Reviewer's 1 comments

General Comment: Thank you very much for giving me the opportunity to review this work. Below I present some questions with the aim of helping to improve the presentation of the work carried out.

Response #: Thank you for your positive comment.

CONSIDERATIONS FOR AUTHORS

Comment #1: Line 281-294. The sample size to be included in the study is not specified.

Response #1: Thank you for pointing out this. We have included the sample size estimation, please see page 8-9, lines 349-360:

“Our study evaluates the effects of a holistic rehabilitation program on individuals with MS, utilizing key metrics to determine improvements in functional ability, nutritional status, and psychological well-being. Determining the minimum required sample size for our protocol was guided by preliminary data on quality of life (29). Sample size calculations were performed using G*Power 3.0 (30), employing the “ANOVA: Repeated measures, within-between interaction” approach. This method facilitated estimating the study's power to detect between-group differences and within-subject changes over time, considering three measurement points (baseline, mid-intervention, and post-intervention) for our MS participants. We set the statistical power and α error probability at 0.90 and 0.05, respectively, aligning with rigorous research standards to ensure sufficient sensitivity in detecting meaningful effects. Our calculations indicate that each group requires 15 individuals to power the study sufficiently.”

Comment #2: Which professional/s carries out the assessment, and how many assessment participate?

Response #2: Thank you for raising this important question. The assessments are conducted by a multidisciplinary team of professionals, ensuring a comprehensive evaluation of the intervention's impact. In response to your comment, we have added the text below, please see page 3-4, lines 130-133.

“To ensure a comprehensive evaluation of the intervention's impact, a team of professionals from different disciplines, including clinical exercise physiology, nutrition, psychology, and environmental physiology, conduct the assessments.”

Comment #3: Line 264. Intervention Control Group. The control group continues with the usual treatments, does the intervention group also or are they suspended?

Response #3: Thank you for your insightful comment. In our study, the control group continues with their usual care treatments throughout the duration of the trial. This approach ensures that we maintain a standard level of care, reflecting the typical management strategies for individuals with Multiple Sclerosis in a non-experimental context. Enrolled participants may continue with their usual care (e.g. attend physical or occupational therapy sessions) but are not allowed to participate in a structured exercise program or psychological therapy during the intervention. We have specified this at page 3, lines 133-142:

‘Enrolled participants are encouraged to continue with their usual care, which may vary based on factors such as disability status and disease severity. Usual care often includes services such as disease-modifying treatments, various aids, and physical or occupational therapy, all of which are essential for managing their condition and ensuring optimal support. However, to prevent potential confounding effects on the study outcomes, participants are advised to refrain from participating in structured exercise programs or psychological therapy during the intervention period. This ensures that any changes observed in the study are attributed specifically to the intervention being tested, rather than to additional interventions that participants may undergo concurrently.’

Comment #4: What do exercise programs include and what recommendations are given to participants? It would be convenient to know the program that the participants must follow and if modifications are made, based on what?

Response #4: Thank you for your feedback. We have addressed all the points you raised with a detailed description in the manuscript (page 6, lines 230-272)

“The exercise program included in this intervention has been specifically designed to encourage participants to increase physical activity while reducing sedentary behavior, promoting behavior change. Participants will engage in structured exercise sessions that are tailored to promote activity behavior change, which may include a variety of exercises such as aerobic activities (e.g., walking, cycling), strength training, flexibility exercises, and balance training, depending on the participant's goals and capabilities. An exercise physiologist will provide regular support and guidance to ensure that participants are on track. The exercise program will include:

Education Sessions: Participants will meet virtually with an exercise physiologist every two weeks via Microsoft Teams. During these sessions, participants will receive per-sonalized education about their specific needs, including exercise techniques, goal set-ting, overcoming barriers to physical activity, and strategies for increasing daily movement and reducing sedentary time. Furthermore, the exercise physiologist will discuss the benefits of regular exercise. These benefits may include improved cardiovascular health, enhanced muscular strength and endurance, better flexibility and balance, increased energy levels, stress reduction, mood enhancement, weight management, and overall improvement in quality of life. Understanding these benefits will help participants appreciate the importance of incorporating exercise into their daily routine and motivate them to adhere to the program.

Exercise Videos: Weekly, customized exercise videos will be uploaded to the Teams platform for participants to access. These videos will provide guidance on performing exercises safely and effectively, allowing participants to engage in physical activity at their convenience while receiving ongoing support and instruction.

Mobile App Integration: A mobile app will be developed to track and monitor participants' daily exercise levels. The app will be integrated with Google Health Fit and iOS Health Fit, allowing participants to monitor their progress, set goals, receive reminders easily, and access resources to support their physical activity progress.

The exercise program will be tailored to each participant's needs, abilities, and preferences. Modifications may be made based on factors such as fitness level, health status, mobility limitations, and personal goals. Throughout the intervention period, participants' progress will be closely monitored through functional capacity tests, in-cluding the 2-minute walk test, the five times sit-to-stand test e.t.c. These tests will be conducted every three weeks to assess participants' functional abilities and overall im-provement. Based on the results of these functional capacity tests and ongoing moni-toring of participants' progress, the intensity and content of the exercise program will be adjusted accordingly. If participants demonstrate improvements in their functional capacity and overall fitness levels, the exercise physiologists will modify the program to gradually increase the intensity or introduce new exercises to continue challenging them. Conversely, if participants experience difficulties or limitations during the program, modifications will be made to ensure their safety and comfort while promoting progress toward their goals. These modifications may include reducing the intensity of certain exercises, focusing on specific areas of weakness or discomfort, or providing alternative exercises that better suit the individual's needs.”

Comment #5: Nutrition education, what is it based on? How are changes measured?

Response #5: Thank you for raising this important question. We have revised the text to address your valuable comments (page 7, lines 274-293)

Nutrition education within the intervention is based on evidence-based principles of maintaining a balanced diet to support the specific needs of individuals with the condition. The guidance provided by the nutritionist is tailored to each participant's dietary requirements, taking into account factors such as their medical history, nutritional needs, and personal preferences. Changes in participants' dietary habits and adherence to nutritional recommendations are measured through various methods. These may include dietary assessments such as food diaries or 24-hour recall interviews to track participants' food intake and identify any changes over time. Additionally, participants' weight, body composition, and other relevant health markers may be monitored to assess the impact of dietary changes on their overall health and well-being. A nutritious recipe is uploaded to the platform each week to further support participants in adopting healthy eating habits. These recipes not only provide practical guidance on preparing nutritious meals but also serve as a source of inspiration and motivation for participants to incorporate healthier food choices into their daily lives.

Overall, the nutrition education component of the intervention aims to empower participants with the knowledge and skills they need to make informed dietary choices and optimize their nutritional intake to support their condition and overall health. Regular meetings with the nutritionist and access to practical resources such as nutritious recipes enhance participants' ability to adhere to dietary recommendations and achieve their nutritional goals.

Comment #6: In Psychotherapy, if the approaches are done in groups, how are the groups organized?

Response #6: After carefully reviewing and addressing all of the concerns you raised, we have revised this section, please see page 7, Line 295-309.

In the psychological component of the intervention, group organization is carefully tailored to ensure participants benefit optimally from shared experiences and support. Led by a psychologist, weekly group meetings conducted via Microsoft Teams focus on addressing participants' psychological well-being. These sessions utilize a combination of Acceptance and Commitment Therapy (ACT) techniques rooted in the tradition of behavioral therapy. Group organization may be based on factors such as participant preference or similarity in MS symptoms and challenges. ACT emphasizes psychological flexibility as a critical process for improving mental health, with core components including cognitive defusion, acceptance, self as context, values clarification, and committed action. Additionally, participants receive supplementary psychoeducational reading material aimed at enhancing coping strategies for managing MS challenges. This material provides valuable insights and practical tools to help participants manage MS more effectively. By integrating group therapy sessions with ACT techniques and psychoeducational resources, participants are equipped with a comprehensive toolkit to manage MS symptoms, negotiate challenges, and improve their overall quality of life (27).

Comment #7: The cooling strategy, how is it designed? How long maximum and how much minimum? What temperature? Does it matter if it is applied in the morning or afternoon?

Response #7: We have updated the cooling strategy description to address your comments. Please see Page 7-8, Lines 311-322.

“The cooling strategy is designed to address temperature regulation in MS and provide symptomatic relief for participants experiencing thermo-sensitivity. Participants will be advised to apply a cooling cap (Headcool Power, Inuteq) to the head and neck area for a minimum of two hours daily. This cooling cap has been chosen for its effectiveness in providing targeted cooling and alleviating MS-related symptoms associated with heat sensitivity. While the study does not specify a maximum duration for the cooling strategy, the minimum duration of two hours daily is recommended to achieve optimal benefits. Regarding temperature, the cooling cap is designed to maintain a consistent cooling temperature within a comfortable range suitable for individuals with MS. The specific time of day when the cooling will be applied is not critical. However, participants will be advised to use the cooling method during their exercise sessions, when engaged in outdoor activities, and any time they do physical work that may increase their body temperature.”

Comment #8: Who makes, and how, the distribution of patients to the control group or intervention group?

Response #8: Thank you for bringing this to our attention. We have addressed it on page 3, line 117-119.

“After baseline testing, the principal investigator will perform randomization using sealed envelopes to allocate participants in a 1:1 ratio to either the BeCool or Control group.”

Reviewer 2 Report

Comments and Suggestions for Authors

This paper is a study protocol of a RCT on a holistic rehabilitation program. However, there are certain limitations. Please see my comments below:

The introduction is extremely long. Please shorten to 2-3 paragraphs.

Methods

Eligibility – does phenotype of MS matter? Age? 

‘have not used 109 cooling therapy for at least 4 months prior to participation in the protocol. ‘is extremely subjective. Most pwMS have some level of heat sensitivity, where their symptoms flare up in heat, and would have used some kind of cooling. Either expand on this or exclude this criteria.

‘Additionally, individuals with medical conditions such as diabetes 112 mellitus, hypertension, heart disease, or kidney disease, and those who are taking medi- 113 cations to manage MS-related symptoms such as antidepressants, psychostimulants, an- 114 ticonvulsants, antispasmodic, and anticholinergic drugs, will not be eligible for the study. ‘ Why not? Many pwMS are on symptomatic management, and these pwMS would benefit from a holistic program. Same goes for this exclusion criteria ‘Furthermore, in- 117 dividuals who have received formal training in mindfulness methods or are currently 118 practicing meditation will not be eligible. ‘

You will need to account for baseline demographics and MS disease modifying therapy. 

Outcome – what is the primary outcome? These are many outcomes described. Also, what about the EDSS? That is usually a key outcome measure. Another outcome measure that should be considered in the study is the PDDS (patient determined disability) – please see Foong et al, The Patient-Determined Disease Steps scale is not interchangeable with the Expanded Disease Status Scale in mild to moderate multiple sclerosis, EJN 2024, https://onlinelibrary.wiley.com/doi/10.1111/ene.16046

The intervention is poorly described. Section 2.7.4 does not make sense ‘2.7.4. Head and Neck Cooling Strategy 252 

Recognizing the importance of temperature regulation in MS, participants will be 253 advised to apply a cooling strategy to the head and neck area for at least two hours daily, 254 providing symptomatic relief. A cooling cap (Headcool Power, Inuteq) The study will not 255 include participants who do not exhibit thermo-sensitivity as per the exclusion criteria. 256 Cool program. ‘. Do you really think pwMS will wear a cooling cap for 2 hours every day? Consider summarizing these key interventions in a table. What is the costing of the program? Is the nutritionist review in person or online?

The lack of a proper control is a major issue. Ideally it should be compared against another rehabilitation program. See https://link.springer.com/article/10.1007/s10072-022-06167-9

Section 3.1 – so what is the actual sample size? 

Comments on the Quality of English Language

-

Author Response

Point to Point Response to Reviewer’s 2 comments.

General Comment #: This paper is a study protocol of a RCT on a holistic rehabilitation program. However, there are certain limitations. Please see my comments below:

Comment #1: The introduction is extremely long. Please shorten it to 2-3 paragraphs.

Response #1: Thank you for your insightful comment regarding the length of the introduction in our manuscript. We understand your concern about its length and the importance of a concise and focused introduction for the paper's overall readability and impact. Accordingly, we have tried to reduced the Introduction, as you suggested. However, it is important to present the background and evidence that leads to this protocol's design. Please see Pages 1-2, Lines 30-96.

Methods

Comment #1: Eligibility – does phenotype of MS matter? Age?

Response #1: Thank you for your insightful comment regarding the eligibility criteria for our study. To address the concerns raised, we have carefully revised the eligibility criteria to ensure that they align with the goals of the study and facilitate the inclusion of a homogeneous study population. We have modified the eligibility criteria to address your comment, please see Page 3, Lines 103-110.

“Prospective participants will be eligible for the study if they meet the following criteria: they have a confirmed diagnosis of MS according to the McDonald criteria (20), exhibit the Relapsing-Remitting MS (RRMS) phenotype characterized by clearly defined attacks of new or increasing neurological symptoms, have experienced no relapses in the past six months, are between the ages of 20 and 60 years, have a confirmed diagnosis of MS for at least 2 years, and exhibit mild or moderate neurological disability, as defined by scoring between 0 to 5.5 on the Expanded Disability Status Scale (EDSS), indicating their ability to walk independently for at least 100 meters without a cane (21).”

Comment #2: ‘have not used 109 cooling therapy for at least 4 months prior to participation in the protocol. ‘is extremely subjective. Most pwMS have some level of heat sensitivity, where their symptoms flare up in heat, and would have used some kind of cooling. Either expand on this or exclude this criteria.

Response #2: We appreciate the reviewer's feedback regarding the eligibility criterion related to the use of cooling therapy. To address the reviewer comment we have exclude this criterion.

Comment #3: ‘Additionally, individuals with medical conditions such as diabetes mellitus, hypertension, heart disease, or kidney disease, and those who are taking medications to manage MS-related symptoms such as antidepressants, psychostimulants, anticonvulsants, antispasmodic, and anticholinergic drugs, will not be eligible for the study. ‘ Why not? Many pwMS are on symptomatic management, and these pwMS would benefit from a holistic program. Same goes for this exclusion criteria ‘Furthermore, individuals who have received formal training in mindfulness methods or are currently 118 practicing meditation will not be eligible. ‘

Response #3: Thank you for raising this concern about the eligibility criteria. To address the reviewer comments, we have excluded these criteria and modified the text accordingly. Please see Page 3, Lines 111-119

“The study will not include participants who do not exhibit thermo-sensitivity as per the exclusion criteria. Additionally, individuals who have undergone psychotherapy in the past six months will also be excluded. Those with severe suicidality, including ideation, plan, and intent, or have experienced one or more relapses in the previous month, or have undergone corticosteroid treatment in the last month, or have other severe medical conditions in addition to MS, or are currently pregnant, will not be able to participate in the study.”

Comment #4: You will need to account for baseline demographics and MS disease modifying therapy.

Response #4: We appreciate the reviewer's feedback regarding the need to account for baseline demographics and MS disease-modifying therapy (DMT) in our study. We completely agree with reviewer comment. To address this concern, we have added the text below, please see Page 9, Lines 364-375

“Detailed documentation and analysis of baseline demographics, including age, gender, disease duration, and relevant medical history, will provide valuable insights into the characteristics of our study population and allow us to control for potential confounding variables in our analyses. Additionally, thorough documentation of participants' current disease-modifying therapy (DMT) regimens, including the type of medication, dosage, and duration of treatment, will enable us to assess the impact of DMT use on intervention outcomes and explore potential interactions with our intervention. We aim to conduct subgroup analyses to investigate differences in treatment effects based on baseline demographics and DMT use, with the goal of informing personalized approaches to MS management. By implementing these strategies, we aim to enhance the validity and generalizability of our findings and ultimately improve the care and outcomes of individuals living with MS.”

Comment #5: Outcome – what is the primary outcome? These are many outcomes described. Also, what about the EDSS? That is usually a key outcome measure. Another outcome measure that should be considered in the study is the PDDS (patient determined disability) – please see Foong et al, The Patient-Determined Disease Steps scale is not interchangeable with the Expanded Disease Status Scale in mild to moderate multiple sclerosis, EJN 2024, https://onlinelibrary.wiley.com/doi/10.1111/ene.16046

Response #5: Thank you for raising this concern about. We have taken your comment into account and added the measurement that you have suggested, please see Page 6, Lines .221-224.

“8. The patient-determined disability scale (PDDS): The PDDS serves as a valuable self-reported outcome measure for assessing disability. This scale allows patients to provide their perspectives on the impact of MS on their daily functioning, reflecting their unique experiences and challenges (27).”

Comment #6: The intervention is poorly described. Section 2.7.4 does not make sense ‘2.7.4. Head and Neck Cooling Strategy 252

Response #6: Thank you for your comment. We have provided a detailed description in this section. Please refer to Page 6, Lines 225- 322.

“2.5 Intervention

This program is structured to provide a holistic, supportive, and adaptable ap-proach to managing MS, focusing on physical health, nutritional balance, psychological support, and practical strategies for daily living (Figure 1). The program includes:

2.5.1 Exercise Intervention and Education

The exercise program included in this intervention has been specifically designed to encourage participants to increase physical activity while reducing sedentary behavior, promoting behavior change. Participants will engage in structured exercise sessions that are tailored to promote activity behavior change, which may include a variety of exercises such as aerobic activities (e.g., walking, cycling), strength training, flexibility exercises, and balance training, depending on the participant's goals and capabilities. An exercise physiologist will provide regular support and guidance to ensure that participants are on track. The exercise program will include:

Education Sessions: Participants will meet virtually with an exercise physiologist every two weeks via Microsoft Teams. During these sessions, participants will receive personalized education about their specific needs, including exercise techniques, goal setting, overcoming barriers to physical activity, and strategies for increasing daily movement and reducing sedentary time. Furthermore, the exercise physiologist will discuss the benefits of regular exercise. These benefits may include improved cardiovascular health, enhanced muscular strength and endurance, better flexibility and balance, increased energy levels, stress reduction, mood enhancement, weight management, and overall improvement in quality of life. Understanding these benefits will help participants appreciate the importance of incorporating exercise into their daily routine and motivate them to adhere to the program.

Exercise Videos: Weekly, customized exercise videos will be uploaded to the Teams platform for participants to access. These videos will provide guidance on performing exercises safely and effectively, allowing participants to engage in physical activity at their convenience while receiving ongoing support and instruction.

Mobile App Integration: A mobile app will be developed to track and monitor participants' daily exercise levels. The app will be integrated with Google Health Fit and iOS Health Fit, allowing participants to monitor their progress, set goals, receive reminders easily, and access resources to support their physical activity progress.

The exercise program will be tailored to each participant's needs, abilities, and preferences. Modifications may be made based on factors such as fitness level, health status, mobility limitations, and personal goals. Throughout the intervention period, participants' progress will be closely monitored through functional capacity tests, in-cluding the 2-minute walk test, the five times sit-to-stand test e.t.c. These tests will be conducted every three weeks to assess participants' functional abilities and overall im-provement. Based on the results of these functional capacity tests and ongoing moni-toring of participants' progress, the intensity and content of the exercise program will be adjusted accordingly. If participants demonstrate improvements in their func-tional capacity and overall fitness levels, the exercise physiologists will modify the program to gradually increase the intensity or introduce new exercises to continue challenging them. Conversely, if participants experience difficulties or limitations during the program, modifications will be made to ensure their safety and comfort while promoting progress toward their goals. These modifications may include reduc-ing the intensity of certain exercises, focusing on specific areas of weakness or dis-comfort, or providing alternative exercises that better suit the individual's needs.

2.5.2 Nutritional Evaluation and Education

Nutrition education within the intervention is based on evidence-based principles of maintaining a balanced diet to support the specific needs of individuals with the condition. The guidance provided by the nutritionist is tailored to each participant's dietary requirements, taking into account factors such as their medical history, nutri-tional needs, and personal preferences. Changes in participants' dietary habits and ad-herence to nutritional recommendations are measured through various methods. These may include dietary assessments such as food diaries or 24-hour recall interviews to track participants' food intake and identify any changes over time. Additionally, par-ticipants' weight, body composition, and other relevant health markers may be moni-tored to assess the impact of dietary changes on their overall health and well-being. A nutritious recipe is uploaded to the platform each week to further support participants in adopting healthy eating habits. These recipes not only provide practical guidance on preparing nutritious meals but also serve as a source of inspiration and motivation for participants to incorporate healthier food choices into their daily lives.

Overall, the nutrition education component of the intervention aims to empower participants with the knowledge and skills they need to make informed dietary choices and optimize their nutritional intake to support their condition and overall health. Regular meetings with the nutritionist and access to practical resources such as nutri-tious recipes enhance participants' ability to adhere to dietary recommendations and achieve their nutritional goals.

2.5.3 Psychological Intervention

In the psychological component of the intervention, group organization is carefully tailored to ensure participants benefit optimally from shared experiences and support. Led by a psychologist, weekly group meetings conducted via Microsoft Teams focus on addressing participants' psychological well-being. These sessions utilize a combination of Acceptance and Commitment Therapy (ACT) techniques rooted in the tradition of behavioral therapy. Group organization may be based on factors such as participant preference or similarity in MS symptoms and challenges. ACT emphasizes psycholog-ical flexibility as a critical process for improving mental health, with core components including cognitive defusion, acceptance, self as context, values clarification, and committed action. Additionally, participants receive supplementary psychoeducational reading material aimed at enhancing coping strategies for managing MS challenges. This material provides valuable insights and practical tools to help participants manage MS more effectively. By integrating group therapy sessions with ACT techniques and psychoeducational resources, participants are equipped with a comprehensive toolkit to manage MS symptoms, negotiate challenges, and improve their overall quality of life (28).

2.5.4 Head and Neck Cooling Strategy

The cooling strategy is designed to address temperature regulation in MS and provide symptomatic relief for participants experiencing thermo-sensitivity. Participants will be advised to apply a cooling cap (Headcool Power, Inuteq) to the head and neck area for a minimum of two hours daily. This cooling cap has been chosen for its effectiveness in providing targeted cooling and alleviating MS-related symptoms associated with heat sensitivity. While the study does not specify a maximum duration for the cooling strategy, the minimum duration of two hours daily is recommended to achieve optimal benefits. Regarding temperature, the cooling cap is designed to main-tain a consistent cooling temperature within a comfortable range suitable for individuals with MS. The specific time of day when the cooling will be applied is not critical. However, participants will be advised to use the cooling method during their exercise sessions, when engaged in outdoor activities, and any time they do physical work that may increase their body temperature.”

Comment #7: Recognizing the importance of temperature regulation in MS, participants will be 253 advised to apply a cooling strategy to the head and neck area for at least two hours daily, 254 providing symptomatic relief. A cooling cap (Headcool Power, Inuteq) The study will not 255 include participants who do not exhibit thermo-sensitivity as per the exclusion criteria. 256 Cool program. ‘. Do you really think pwMS will wear a cooling cap for 2 hours every day? Consider summarizing these key interventions in a table. What is the costing of the program? Is the nutritionist review in person or online?

Response #7: Thank you for your valuable feedback and insightful questions. We have modified the text to address your comments (Page 7-8, Lines 311-322). The nutrition sessions will be conducted online, while all evaluations will be held on site. The program cost cannot be estimated at this time.

Comment #8: The lack of a proper control is a major issue. Ideally it should be compared against another rehabilitation program. See https://link.springer.com/article/10.1007/s10072-022-06167-9.

Response #8: Thank you for raising this concern about the need to have an active control group Accordingly we have modified the text in the paper. Please see Page 8, Line 334-337

Throughout the program, participants will be encouraged to elevate their habitual levels of exercise. This encouragement will be facilitated by the utilization of a newly developed application designed to estimate participants' exercise levels through activity tracking.

Comment #9: Section 3.1 – so what is the actual sample size?

Response #9: Thank you for bringing this to our attention. We have added the following text., Page 8-9, Lines 349-360.

Our study evaluates the effects of a holistic rehabilitation program on individuals with MS, utilizing key metrics to determine improvements in functional ability, nutri-tional status, and psychological well-being. Determining the minimum required sample size for our protocol was guided by preliminary data on quality of life (30). Sample size calculations were performed using G*Power 3.0 (31), employing the “ANOVA: Repeated measures, within-between interaction” approach. This method facilitated estimating the study's power to detect between-group differences and within-subject changes over time, considering three measurement points (baseline, mid-intervention, and post-intervention) for our MS participants. We set the statistical power and α error probability at 0.90 and 0.05, respectively, aligning with rigorous research standards to ensure sufficient sensitivity in detecting meaningful effects. Our calculations indicate that each group requires 15 individuals to power the study sufficiently.

Reviewer 3 Report

Comments and Suggestions for Authors

In this protocol, authors propose a multidisciplinary rehabilitation program including  exercise training, nutritional guidance, psychological support and cooling strategy for enhancing the quality of life of individuals with MS.

Major comments

The clinical course of MS is highly variable and the number of participants must take this notion into account. How many participants are planned for this study? Is there any preliminary data from a pilot study in your center to optimize sample size ? Will the study be conducted in a single-centre or multicentre?  

What will be the age of the participants? Will there be as many men as women since MS is more common among women? Will anteriority in the disease be taken into account? Consideration of ongoing treatment to form study groups

 What will be the reference BMI for nutritional monitoring? Line 71 : Incorporating a diet rich in important nutrients, such as vitamin D, biotin, and other vital vitamins and minerals …. This point is interesting but the assessment of nutritional intakes only through questionnaires is not robust enough.

Protocol data analysis takes into account 3 reference times : before,  3 months and 6 months. Is this enough to assess an improvement in quality of life?

Minor comments

Line 255 A cooling cap (Headcool Power, Inuteq)… ″ : incomplete sentence

Line 346 Data Availability Statement: The data that support the findings of this protocol are available  the corresponding author upon reasonable request″ : the protocol does not refer to this data ?

 Taking into account these remarks will improve the feasibility of this protocol

Author Response

We thank Reviewer 3 for the insightful comment, which has led to the improvement of the paper.

Point to Point response to Reviewer’s 3 comments

In this protocol, authors propose a multidisciplinary rehabilitation program including exercise training, nutritional guidance, psychological support and cooling strategy for enhancing the quality of life of individuals with MS.

Major comments

Comment #1: The clinical course of MS is highly variable and the number of participants must take this notion into account. How many participants are planned for this study? Is there any preliminary data from a pilot study in your center to optimize sample size ? Will the study be conducted in a single-center or multicentre? 

Response #1: Thank you for your insightful comment. We have added the sample size calculation (pages 8-9, Lines 349-360) and mentioned that this study will be a single-center, only the members of the Greek Multiple Sclerosis society will participate (page 3, line 123).

“Our study evaluates the effects of a holistic rehabilitation program on individuals with MS, utilizing key metrics to determine improvements in functional ability, nutri-tional status, and psychological well-being. Determining the minimum required sample size for our protocol was guided by preliminary data on quality of life (30). Sample size calculations were performed using G*Power 3.0 (31), employing the “ANOVA: Re-peated measures, within-between interaction” approach. This method facilitated esti-mating the study's power to detect between-group differences and within-subject changes over time, considering three measurement points (baseline, mid-intervention, and post-intervention) for our MS participants. We set the statistical power and α error probability at 0.90 and 0.05, respectively, aligning with rigorous research standards to ensure sufficient sensitivity in detecting meaningful effects. Our calculations indicate that each group requires 15 individuals to power the study sufficiently.”

Comment #2: What will be the age of the participants? Will there be as many men as women since MS is more common among women? Will anteriority in the disease be taken into account? Consideration of ongoing treatment to form study groups.

Response #2: Thank you for your inquiry regarding the characteristics of the study participants. We will aim to recruit participants within a specified age range, typically between 20 and 60 years old, to ensure that we capture a broad representation of individuals affected by MS. In terms of gender representation, while MS does indeed affect more women than men, we will strive to recruit an equal number of male and female participants whenever feasible. This gender balance will allow us to explore potential gender-specific differences in response to interventions and ensure that study findings are applicable to both male and female populations affected by MS. Regarding disease severity and duration, we will consider factors such as disease duration and level of disability, as measured by standardized clinical assessments such as the Expanded Disability Status Scale (EDSS). By stratifying participants based on disease severity and duration, we can better understand how these factors may influence intervention outcomes and tailor treatment approaches accordingly. Moreover, we will carefully consider ongoing MS treatments when forming study groups to ensure that participants receive appropriate care and minimize potential confounding effects on study outcomes. This may involve stratifying participants based on their current disease-modifying therapies (DMTs) or other symptomatic treatments to account for differences in treatment regimens across study groups. We have discussed all of these points at various stages throughout the paper.

Page 3, Lines 103-119

2.1 Eligibility criteria

Prospective participants will be eligible for the study if they meet the following criteria: they have a confirmed diagnosis of MS according to the McDonald criteria (20), exhibit the Relapsing-Remitting MS (RRMS) phenotype characterized by clearly defined attacks of new or increasing neurological symptoms, have experienced no relapses in the past six months, are between the ages of 20 and 60 years, have a confirmed diagnosis of MS for at least 2 years, and exhibit mild or moderate neurological disability, as de-fined by scoring between 0 to 5.5 on the Expanded Disability Status Scale (EDSS), in-dicating their ability to walk independently for at least 100 meters without a cane (21).

The study will not include participants who do not exhibit thermo-sensitivity as per the exclusion criteria. Additionally, individuals who have undergone psychotherapy in the past six months will also be excluded. Those with severe suicidality, including ideation, plan, and intent, or have experienced one or more relapses in the previous month, or have undergone corticosteroid treatment in the last month, or have other severe medical conditions in addition to MS, or are currently pregnant, will not be able to participate in the study. After baseline testing, the principal investigator will perform randomization using sealed envelopes to allocate participants in a 1:1 ratio to either the BeCool or Control group.

Page 9, Lines 364-375

Detailed documentation and analysis of baseline demographics, including age, gender, disease duration, and relevant medical history, will provide valuable insights into the characteristics of our study population and allow us to control for potential confounding variables in our analyses. Additionally, thorough documentation of participants' current disease-modifying therapy (DMT) regimens, including the type of medication, dosage, and duration of treatment, will enable us to assess the impact of DMT use on intervention outcomes and explore potential interactions with our intervention. We aim to conduct subgroup analyses to investigate differences in treatment effects based on baseline demographics and DMT use, with the goal of informing personalized approaches to MS management. By implementing these strategies, we aim to enhance the validity and generalizability of our findings and ultimately improve the care and outcomes of individuals living with MS.

Comment #3:  What will be the reference BMI for nutritional monitoring? Line 71 : ″Incorporating a diet rich in important nutrients, such as vitamin D, biotin, and other vital vitamins and minerals …″. This point is interesting but the assessment of nutritional intakes only through questionnaires is not robust enough.

Response #3: Thank you for your feedback and insightful questions. In response to your inquiry about the reference BMI for nutritional monitoring, we plan to use BMI classification as a reference. Regarding the assessment of nutritional intakes, we acknowledge the limitations of relying solely on questionnaires and agree that a more comprehensive approach is warranted. To address this concern, we will incorporate multiple methods for assessing participants' nutritional status, including dietary recalls, food diaries, and biochemical markers of nutrient status (e.g., blood levels of vitamin D). These complementary approaches will provide a more robust and holistic evaluation of participants' dietary intakes and nutritional status, allowing for a more accurate assessment of the effectiveness of our intervention in promoting optimal nutrition and overall health. Furthermore, we appreciate your interest in the importance of incorporating a diet rich in essential nutrients, such as vitamin D and biotin, into the intervention. To enhance the assessment of participants' nutritional intakes, we will provide guidance on achieving a balanced diet that includes sources of these vital nutrients.

Comment #4: Protocol data analysis takes into account 3 reference times : before,  3 months and 6 months. Is this enough to assess an improvement in quality of life?

Response #4: Thank you for your question regarding the adequacy of three reference times (before, 3 months, and 6 months) for assessing improvements in quality of life in our protocol data analysis. While this timeframe provides valuable insight into short-term changes following the intervention, we acknowledge that assessing long-term improvements in quality of life may require a more extended follow-up period. While our analysis at three reference times allows us to capture initial changes and trends in quality of life, we recognize the importance of extending follow-up assessments beyond the 6-month mark to evaluate sustained improvements over time. Unfortunately, we cannot conduct further evaluations due to funding constraints. However, we plan to expand this protocol in a larger clinical trial in the future to allow for further assessments.

Furthermore, we will consider conducting subgroup analyses to explore whether certain participant characteristics or intervention components contribute to differential outcomes in quality of life improvement over time. This will allow us to identify factors that may enhance the effectiveness of our intervention and tailor future interventions to better meet the needs of individuals living with MS.

Minor comments

Comment #5: Line 255 ″ A cooling cap (Headcool Power, Inuteq)… ″ : incomplete sentence

Response #5: Thank you very much for pointing out this, we have revised the text.

Comment #6: Line 346 ″Data Availability Statement: The data that support the findings of this protocol are available  the corresponding author upon reasonable request″ : the protocol does not refer to this data ?

Taking into account these remarks will improve the feasibility of this protocol.

Response #6: Thank you for bringing this to our attention. However, we are required by the Journal to include this statement. Your insightful comments have greatly contributed to the improvement of the protocol.

Round 2

Reviewer 1 Report

Comments and Suggestions for Authors

Thanks for the modifications introduced.

Author Response

Comment #1: Thanks for the modifications introduced.

Response #1: We express our gratitude to the reviewer for providing valuable feedback that has significantly improved our paper.

Reviewer 2 Report

Comments and Suggestions for Authors

The authors have responded appropriately to the prior comments and amended the manuscript significantly. I would advice against this criterion "The study will not include participants who do not exhibit thermo-sensitivity as per 112 the exclusion criteria. " - as this is very broad, and most pwMS will have some level of thermosensitivity. This could be omitted or modified. I wish them all the best with the study.

Comments on the Quality of English Language

-

Author Response

Comment #1: The authors have responded appropriately to the prior comments and amended the manuscript significantly. I would advice against this criterion "The study will not include participants who do not exhibit thermo-sensitivity as per 112 the exclusion criteria. " - as this is very broad, and most pwMS will have some level of thermosensitivity. This could be omitted or modified. I wish them all the best with the study.

Response #1: Thank you for your feedback. I understand your concern about the broadness of the exclusion criteria regarding thermo-sensitivity. To address this, we have removed this criterion.  Please see Page 3, Lines 112-116

"The study will not include participants who have undergone psychotherapy in the past six months. Additionally, those with severe suicidality, including ideation, plan, and intent, or have experienced one or more relapses in the previous month, or have undergone corticosteroid treatment in the last month, or have other severe medical conditions in addition to MS, or are currently pregnant, will not be able to participate in the study. "

Reviewer 3 Report

Comments and Suggestions for Authors

The authors have considered the various recommendations and provided convincing and appropriate responses

the manuscript is suitable for publication

Author Response

Comment #1: The authors have considered the various recommendations and provided convincing and appropriate responses the manuscript is suitable for publication.

Response #1: Thank you for your thorough review and positive feedback on our manuscript.